# Physicochemical Properties of Cellulose Separators for Lithium Ion Battery: Comparison with Celgard2325

**DOI:** 10.3390/ma12010002

**Published:** 2018-12-20

**Authors:** Jie Sheng, Ruibin Wang, Rendang Yang

**Affiliations:** 1State Key Laboratory of Pulp and Paper Engineering, South China University of Technology, Guangzhou 510640, China; s498952506@163.com; 2School of Materials and Energy, Center of Emerging Material and Technology, Guangdong University of Technology, Guangzhou 510006, China; wang.rb@gdut.edu.cn

**Keywords:** nanofibril membrane, plant fiber, electrolyte wettability, thermal dimensional stability, tensile strength

## Abstract

High electrolyte wettability, thermal dimensional stability, and tensile strength are prerequisites for implementing separators in practical applications. In this study, we report on the discovery of nanofibril membranes derived from various plant fibers commonly used in the papermaking industry, for low cost and higher performances than the commercially available Celgard2325 in regard to the application of separators for lithium-ion batteries. Nanofibril membranes showed water contact angles as low as 18°, negligible size change at a heating temperature of 160 °C for 120 min, and tensile strength up to 137.6 MPa. The homogenization was found to strongly contribute to these improved performances. These findings suggest that the plant fiber-derived nanofibril membranes are anticipated to be promising candidates as separators for lithium-ion batteries.

## 1. Introduction

Today, lithium-ion batteries (LIBs) are one of the most promising and important energy storage technologies. LIBs can not only be used for portable devices like mobile phones, laptops, and digital cameras, but also flexible/wearable electronics, electronic vehicles, and large-scale power sources [1,2,3,4]. All of the above take advantage of the special properties of LIBs: long cycle life, high energy density, low self-discharging, and no memory effect [5,6,7]. The separator is a crucial component of LIBs. Its properties and structure play an indirect but key role in influencing the cell performance, including service life, energy density, power density, and safety [8,9]. Hence, several factors must be considered while choosing suitable separators for LIBs. Thereinto, high electrolyte wettability, thermal dimensional stability, and tensile strength are prerequisites for implementing separators in practical applications.

The separators used in modern LIBs are mainly polyolefin types such as Celgard2325, due to their good chemical stability [10,11]. Nevertheless, the drawbacks of polyolefin separators are also obvious, especially their limited electrolyte wettability and poor thermal dimensional stability, which are adverse to the performance and safety of LIBs. From a practical point of view, cellulose is a promising alternative for the next generation of high-performance/high-safety batteries, due to its outstanding wettability and thermal dimensional stability, along with the reduced production cost and environmental benignancy [11,12,13,14,15].

However, previous results showed that conventional papers seem unsuitable for LIB separators. Their thickness, pore size, and evenness could not meet the requirements because of massive size of ordinary cellulose fibers. Lee et al. obtained good results by using cellulose nanofibrils [16]. Since then, other studies evaluating a cellulose/polysulfonamide composite membrane and a polydopamine-coated cellulose membrane have also achieved good results [17,18]. However, the tensile strength of all of the aforementioned cellulose membranes is less than half of commercial separators, resulting in difficult assembly and poor anti-dropping capability. For example, the maximum stress of the cellulose/polysulfonamide composite membrane was approximately 18 MPa, while that of Celgard2325 was more than 100 MPa [18]. It is also important to note that most studies used only cotton pulp as raw material. However, the source of cellulose is abundant, and the performance of paper prepared by different cellulose raw materials is quite different. Therefore, exploring the basic properties of different cellulose membranes is significant for promoting the sustainable development of LIBs [19,20].

In this work, the properties of separators derived from various types of plant fibers commonly used in paper making were compared with those of a commercially available separator (i.e., Celgard2325). All separators were prepared via a facile papermaking process, which are used in the large-scale, low-cost production of cellulose LIB separators. It was demonstrated that the plant fiber-derived nanofibril membranes possessed excellent electrolyte wettability, thermal dimensional stability, and tensile strength, which suggested that these plant fiber-derived nanofibril membranes were anticipated to be promising candidates as separators for LIBs.

## 2. Materials and Methods

### 2.1. Materials

Bamboo pulp, hardwood pulp, softwood pulp, cotton pulp, and hemp pulp (Zhejiang Yongtai Co., Hangzhou, China), 1 M lithium hexafluorophosphate (LiPF_6_) in ethylene carbonate (EC)/dimethyl carbonate (DMC)/ethyl methyl carbonate (EMC) (1/1/1, *v*/*v*/*v*) (Nanjing Mojiesi Energy Technology Co., Nanjing, China), and polypropylene (PP)/polyethylene (PE)/polypropylene (PP) separator (Celgard2325) (Shenzhen Kejing Co., Shenzhen, China) were received from their respectively indicated suppliers.

### 2.2. Preparation of the Cellulose Nanofibril Membrane

The formation of the nanocellulose membrane is illustrated in Figure 1. In brief, each pulp source was dispersed to form a suspension with a concentration of 10 wt %, followed by beating in a PFI beater (HAMJERN MASKIN 621, Hamar, Norway) at 20,000 revolutions. Afterward, the resultant suspension was diluted to 1 wt % and subjected to grinding for 40 passes with an ultrafine friction grinding machine (MKZA6-2J, Masuko Sangyo Co., Saitama, Japan). The obtained samples were further diluted to 0.5 wt %, prior to homogenizing (Mini, Noozle, Shanghai, China) for four passes. The homogenized suspensions were then vacuum-filtered through a filtration membrane with pore diameter of 0.45 μm, and each of the obtained filter cakes was sandwiched between two ordinary filter papers with the filtration membrane. Finally, two continuous steps of vacuum drying—(1) 95 °C for 5 min and (2) 100 °C for 24 h—were conducted to form the nanocellulose membranes. For the sake of simplicity, the resulting nanocellulose membranes derived from bamboo pulp, hardwood pulp, softwood pulp, cotton pulp, and hemp pulp were labelled as a2, b2, c2, d2, and e2, respectively. For comparison, nanocellulose membranes without the homogenization were also fabricated, which were correspondingly labeled as a1, b1, c1, d1, and e1, respectively.

### 2.3. Characterizations

The electrolyte wettability of cellulose nanofibril membrane and Celgard2325 was examined by observing the wetted area of the separator surface after dropping the liquid electrolyte onto separators.

The contact angles of deionized water on some selected separators were evaluated using an OCA40 Micro contact angle system (DataPhysics, Filderstadt, Germany). The values were obtained by averaging values from more than five drops.

The thermal shrinkage behavior of cellulose nanofibril membrane and Celgard2325 was determined by an oven test ranging from 80 to 160 °C for 120 min.

The thermal stability of the separator was tested using a differential scanning calorimeter (DSC, TA Q200, New Castle, DE, USA) ranging from 80 to 230 °C at 10 °C min^−1^ under N_2_ atmosphere.

The weight loss as a function of temperature was assessed by means of thermogravimetric analysis (TGA, TA Q500, New Castle, DE, USA). The samples were tested in N_2_ atmosphere with a heating rate of 10 °C min^−1^ in the temperature range of 30–600 °C.

Tensile strength of cellulose nanofibril membrane and Celgard2325 was tested with a universal testing machine (INSTRON 5565, Norwood, MA, USA) at a speed of 5 mm min^−1^ with samples 1.5 cm wide and 3 cm long.

The morphology of the cellulose nanofibril was observed on a scanning electron microscopy (SEM, Quanta FEG 250, Hillsboro, OR, USA).

## 3. Results

### 3.1. Morphology

The plant fiber is roughly cylindrical, the length of the fibers is generally 1–3 mm, and the diameter is mostly 10–20 μm. Treatments like beating, drying, pressing, and calendaring can reduce the length of fibers and the thickness of papers, but to a limited extent. For lightweight and high-energy-density batteries, the thickness of separators should be kept as small as possible. However, the thickness of commercial separators is generally 20–30 μm, which is approximately equivalent to the diameter of ordinary cellulose fibers. At this thickness, the stacking of only a few layers of fiber renders the uniformity and strength of the paper insufficient. Therefore, it is difficult to meet the development of separators by using ordinary cellulose fiber. As shown in Figure 1, a cellulose nanofibril membrane at the same thickness has more fibers, resulting in more uniform pores and higher strength, which is more suitable for preparing high-performance and high-safety LIB separators.

Figure 2 shows the morphology of the cellulose nanofibrils before and after homogenization. As shown in the figure, both before and after homogenization, the diameter of the cellulose nanofibrils reached the nanometer level, mainly distributed in the range of 10–50 nm, and their lengths were in the micrometer level. There was a big difference in nanofibril morphology before and after homogenization. Without homogenization, the nanofibrils were poorly dispersed, and several slender nanofibrils were adhered together. However, after homogenization, a large number of nanofibrils were dispersed into a single fiber. This process contributed to the wettability and tensile strength of cellulose membranes. Since the nanofibrils were prepared by a simple mechanical method, the surface of the nanofibrils was not smooth, and the nanofibrils were visibly twisted. Even at ultra-low concentrations, it was not easy to disperse the cellulose nanofibrils into individual fibers after the homogenization.

### 3.2. Wettability

Figure 3 shows the wettability of an electrolyte for separators prepared from different fiber slurries. As can be seen from the figure, the wettability of the electrolyte to those cellulose membranes was quite different. After being processed by the ultrafine friction grinding machine 40 times, the hardwood pulp and hemp pulp membrane were more easily wetted by the electrolyte, and membranes prepared by the bamboo pulp, softwood pulp, and cotton pulp were relatively poor. After further processing by the homogenizer for three times, the results were similar to those before the treatment, and the hardwood pulp and hemp pulp membrane were still better. By comparing the wetted area, we found that the wettability of the separators after homogenization was improved to some extent. The possible reason is that the fiber dispersion was more uniform and more lyophilic groups were exposed via homogenization. Moreover, compared with the commercial Celgard2325 in Figure 4, the wettability of cellulose nanofibril membranes before and after the homogenization was superior to that of Celgard2325.

A surface contact angle test can more accurately indicate the wettability of each separator. The results are shown in Figure 4 and Figure 5. Since the wettability of the electrolyte to cellulose membranes is too good, the electrolyte is absorbed at the moment of contact with the membrane, without a relatively stable interface. In this experiment, distilled water was used instead of the electrolyte. The results showed that the contact angle of cellulose membranes was less than 45°, which was much smaller than the 86° of Celgard2325. Before and after the homogenization, the hardwood pulp membrane (28°/21°) and the hemp pulp membrane (27°/18°) were the best. Softwood pulp membrane (38°/25°) and cotton pulp membrane (40°/26°) were the second, followed by bamboo pulp membrane (44°/37°), consistent with the previous result. Therefore, hemp pulp and hardwood pulp were suitable for the coating of composite LIB separators, in order to improve the wettability of polyolefin separators.

### 3.3. Thermal Stability

Good thermal dimensional stability of the separator is critical to the safety of batteries. Figure 6 shows the thermal dimensional stability of different cellulose membranes and Celgard2325. As shown in the figure, Celgard2325 maintained a good shape after being placed at 80 °C for 120 min, but shrunk to approximately 92% of the original size at 100 °C, 72% at 120 °C, 64% at 140 °C, 59% at 160 °C, and turned from white to transparent at 140 °C. In contrast, the cellulose membranes maintained a good shape regardless of whether the pulp was homogenized or not, and did not change significantly with the increase in temperature. This indicated that the cellulose membranes had much better thermal dimensional stability than Celgard2325.

Figure 7a shows the differential scanning calorimetry (DSC) curves of Celgard2325 and different cellulose membranes without homogenization. The heating flow curves of the cellulose membranes were relatively smooth in the range of 80 to 230 °C. There was no obvious endothermic or exothermic peak indicative of the stable cellulose membranes in this temperature range. Celgard2325 exhibited distinct endothermic peaks at 133 °C and 159 °C, which were the melting temperatures of PE and PP, respectively. The melting temperature of PE was 133 °C, indicating that it was a high-density polyethylene. Even though the melting temperature of Celgard2325 first appeared at 133 °C, Celgard2325 could not maintain a good shape at 100 °C from the preceding discussion. Therefore, DSC could not fully reflect the thermal stability of the separator alone.

TGA of Celgard2325 and the cellulose membranes with homogenization are compared in Figure 7b. The thermogravimetric curves of the cellulose membranes were similar, with a slight weight loss before 120 °C due to the evaporation of water. The thermal decomposition temperature of each cellulose membrane began at approximately 295 °C, and the decomposition was essentially completed at approximately 370 °C. In contrast, Celgard2325 had a higher thermal decomposition temperature, starting at 410 °C and completing at 470 °C, with a weight loss of approximately 96%. The results showed that the thermal decomposition temperature was not positively correlated with the thermal dimensional stability, and the cellulose membranes with better thermal dimensional stability were more suitable for LIBs than Celgard2325.

### 3.4. Mechanical Properties

Figure 8a shows the stress–strain curves of the cellulose membranes without homogenization. The results showed that the softwood pulp membrane, hardwood pulp membrane, and cotton pulp membrane had good tensile strengths, which were 62.8, 59.6, and 58.3 MPa, respectively. That of the bamboo pulp membrane was poor (i.e., 45.7 MPa), and that of the hemp pulp membrane was the worst, at 32.8 MPa. Due to the vacuum filtration method, these cellulose membranes do not distinguish between the longitudinal and transverse tensile strength. Compared with Celgard2325 in Figure 8c, the tensile strength of cellulose membranes was much better than the transverse tensile strength of Celgard2325 (14.0 MPa), but less than 36% of the longitudinal tensile strength of Celgard2325 (174.4 MPa). Meanwhile, the elongation of cellulose membranes without homogenization was approximately 3% to 6%, which was lower than the transverse elongation of Celgard2325 (8.1%), and less than 10% of the longitudinal elongation of Celgard2325 (79.4%).

The stress–strain curves of the cellulose membranes with homogenization are shown in Figure 8b. The results showed that the tensile strength of the cellulose membranes with homogenization improved to some extent, but the elongation changed slightly. Among them, the tensile strength of hemp pulp membrane was increased to 87.5 MPa, the bamboo pulp membrane to 91.7 MPa, the hardwood pulp membrane to 120.3 MPa, the cotton pulp membrane to 132.6 MPa, and the softwood pulp membrane to 137.6 MPa. By homogenization, the tensile strength of each cellulose membrane exceeded 50% of the longitudinal tensile strength of Celgard2325, even though their thickness was close (Table 1). In particular, the cotton pulp and softwood pulp membrane reached more than 75% of Celgard2325, which met the requirement for matrix material of cellulose LIB separators. However, the homogenization failed to improve the elongation of cellulose membranes. The elongation of cellulose membranes after the homogenization was closer, which could be due to the better dispersion and uniformity of the nanofibrils.

## 4. Conclusions

In summary, the separator prepared from plant fiber commonly used in papermaking industry possessed excellent electrolyte wettability, thermal dimensional stability, and tensile strength, suitable for LIBs. The contact angle of the cellulose membranes (distilled water) was less than 45°, and the membranes had no obvious change when they were placed at 160 °C for 120 min. Moreover, the homogenization effectively improved the electrolyte wettability and tensile strength of the cellulose membranes. After the homogenization, the hemp pulp and hardwood pulp cellulose nanofiber (CNF) membranes had the best wettability, with the contact angles only 18° and 21°, which were suitable for coating of composite LIB separators. Meanwhile, the separators made from softwood pulp and cotton pulp had the best tensile strength, exceeding 130 MPa, which were ideally used as the matrix material of cellulose LIB separators.

## Figures and Tables

**Figure 1 materials-12-00002-f001:**
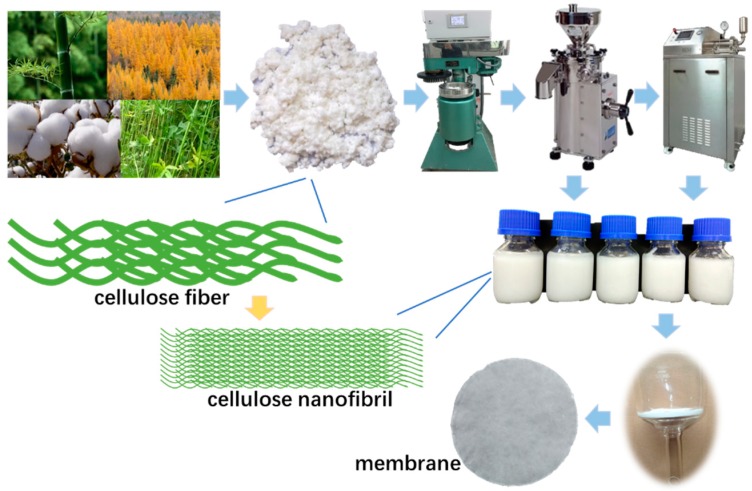
Schematic illustration for the preparation of the cellulose nanofibril membrane.

**Figure 2 materials-12-00002-f002:**
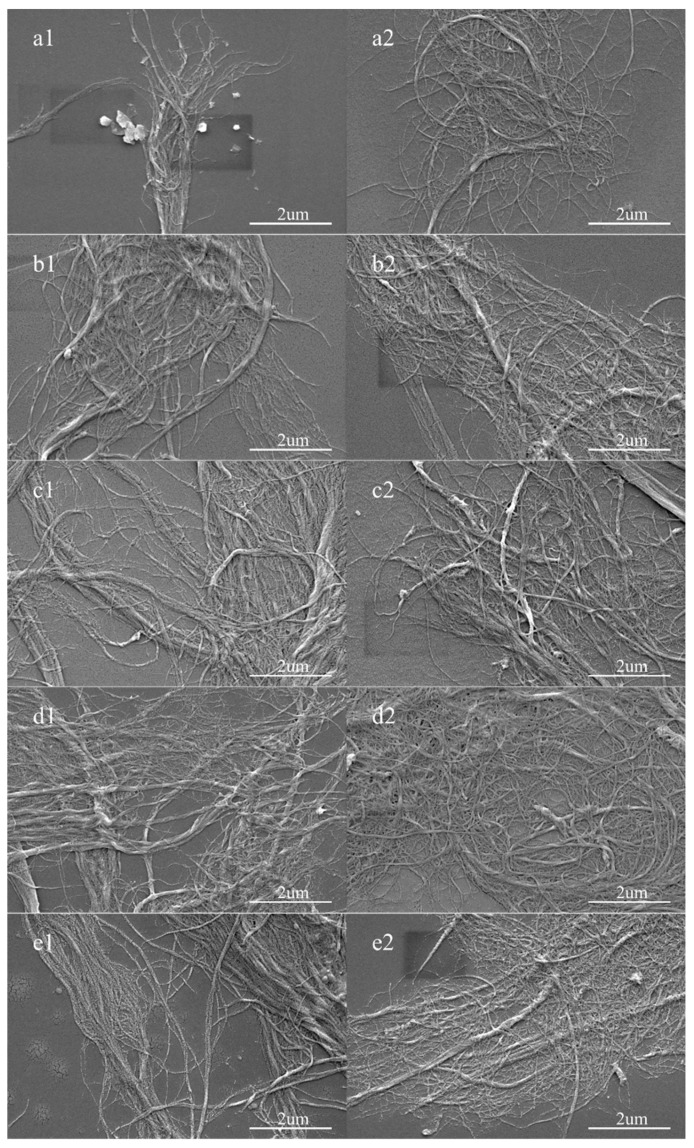
The scanning electron microscope (SEM) images of the cellulose nanofibrils before and after the homogenization: (**a1**,**a2**) bamboo pulp; (**b1**,**b2**) hardwood pulp; (**c1**,**c2**) softwood pulp; (**d1**,**d2**) cotton pulp; (**e1**,**e2**) hemp pulp (figure a2,b2 is taken from Reference [21]).

**Figure 3 materials-12-00002-f003:**
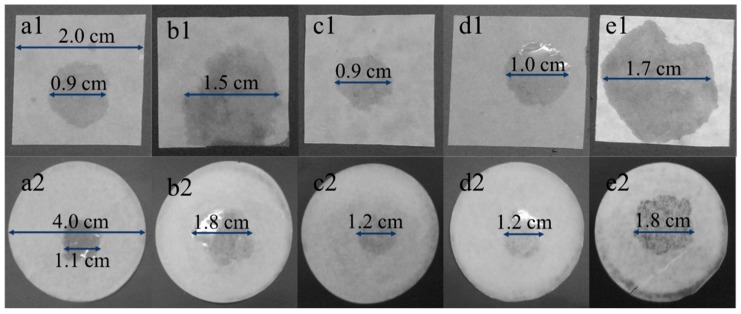
The wetted area of an electrolyte for membranes before and after the homogenization: (**a1**,**a2**) bamboo pulp; (**b1**,**b2**) hardwood pulp; (**c1**,**c2**) softwood pulp; (**d1**,**d2**) cotton pulp; (**e1**,**e2**) hemp pulp.

**Figure 4 materials-12-00002-f004:**
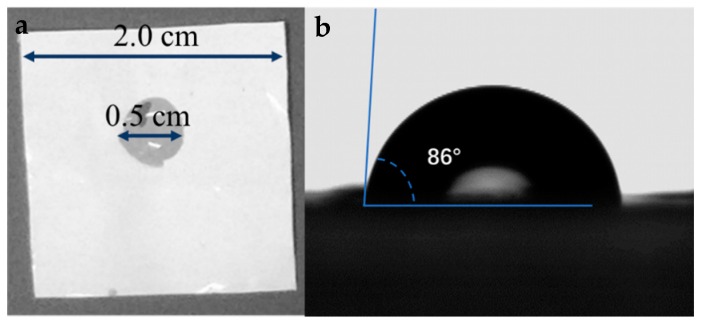
The wettability of Celgard2325: (**a**) the wetted area of an electrolyte and (**b**) the contact angles of water.

**Figure 5 materials-12-00002-f005:**
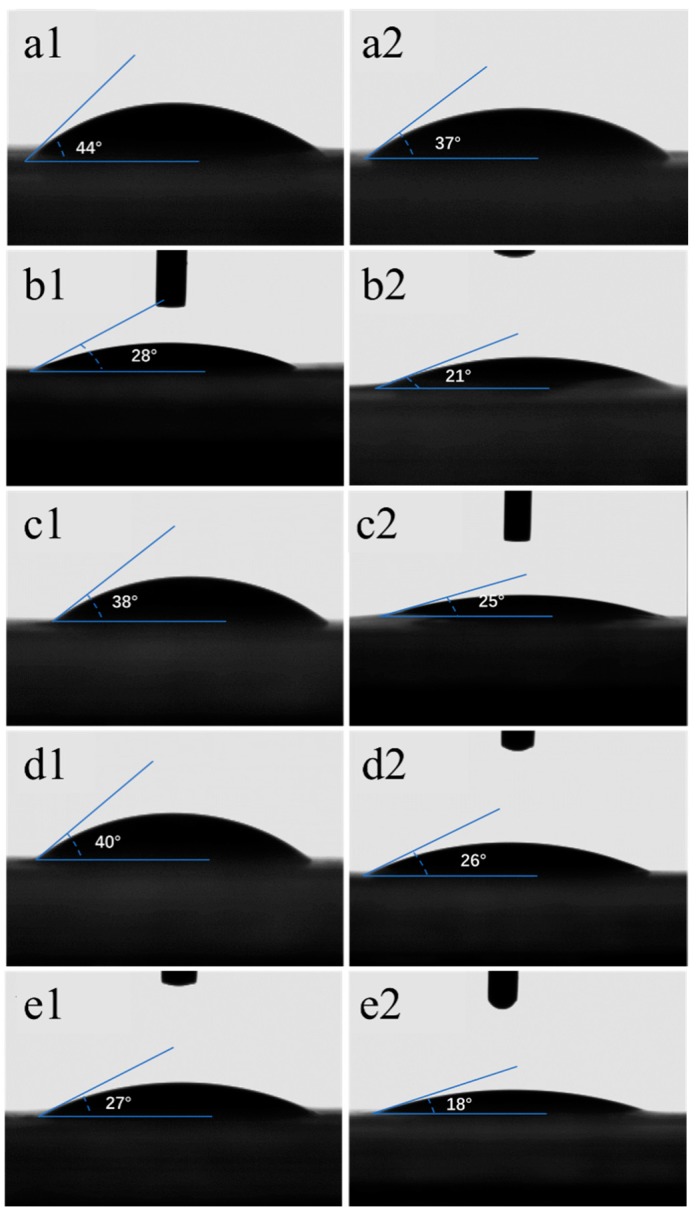
The contact angles of water on the surface of the cellulose membranes before and after the homogenization: (**a1**,**a2**) bamboo pulp; (**b1**,**b2**) hardwood pulp; (**c1**,**c2**) softwood pulp; (**d1**,**d2**) cotton pulp; (**e1**,**e2**) hemp pulp.

**Figure 6 materials-12-00002-f006:**
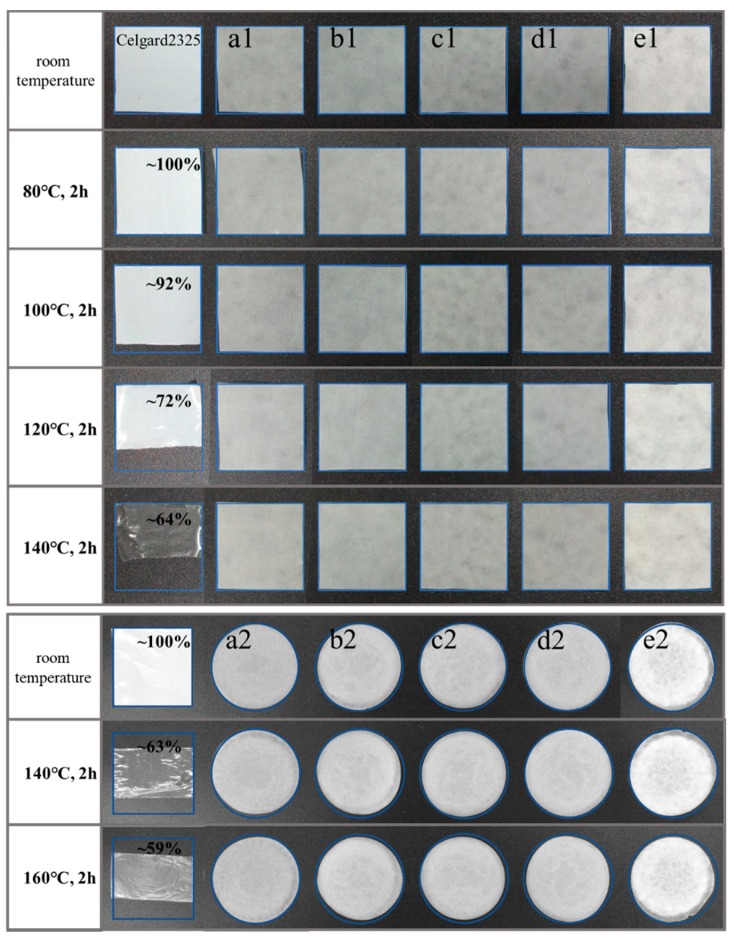
The thermal dimensional stability of Celgard2325 and the cellulose membranes before and after the homogenization: (**a1**,**a2**) bamboo pulp; (**b1**,**b2**) hardwood pulp; (**c1**,**c2**) softwood pulp; (**d1**,**d2**) cotton pulp; (**e1**,**e2**) hemp pulp.

**Figure 7 materials-12-00002-f007:**
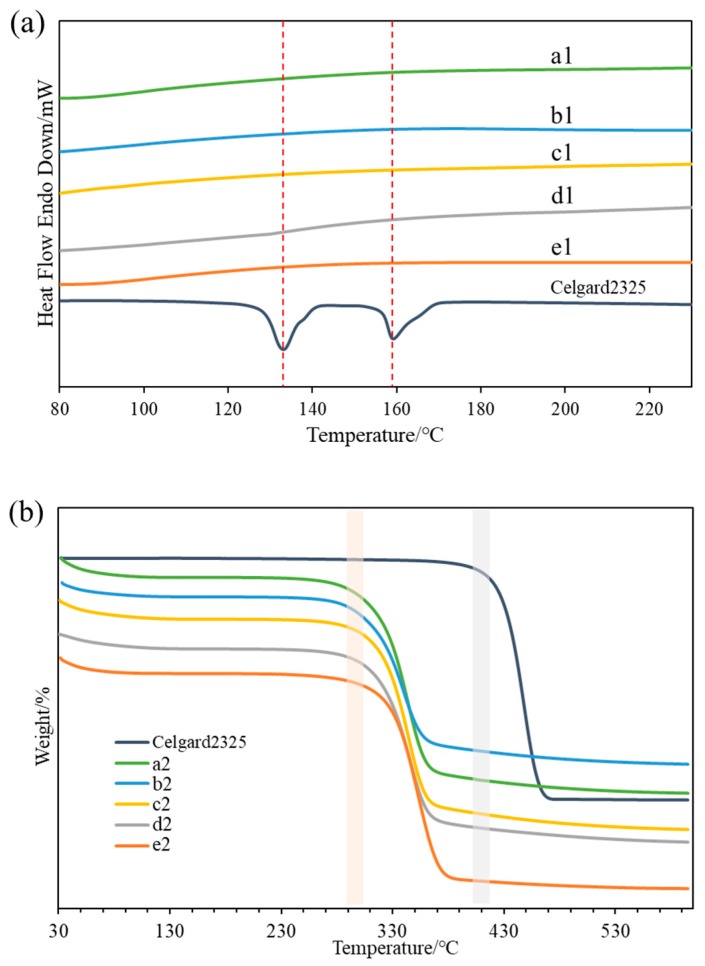
The (**a**) DSC and (**b**) TGA curves of Celgard2325 and the cellulose membranes before and after the homogenization.

**Figure 8 materials-12-00002-f008:**
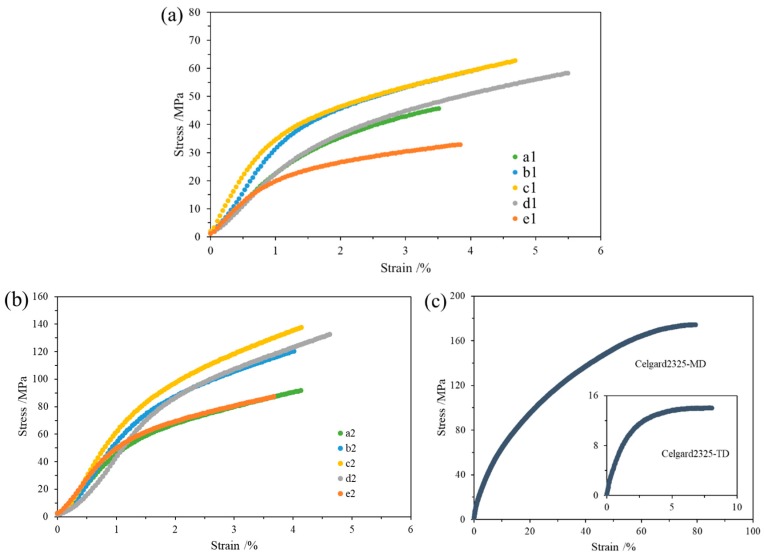
The stress–strain curves of the cellulose membranes (**a**) before and (**b**) after the homogenization, and (**c**) Celgard2325. MD: longitudinal tensile strength; TD: transverse tensile strength.

**Table 1 materials-12-00002-t001:** The thickness of the membranes.

Samples	Thickness (µm)	Samples	Thickness (µm)
a1	35	a2	22
b1	37	b2	21
c1	37	c2	23
d1	38	d2	22
e1	38	e2	22
Celgard2325	24	-	-

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
