# Peer review of "Physicochemical Properties of Cellulose Separators for Lithium Ion Battery: Comparison with Celgard2325"

_materials, 2018, doi:10.3390/ma12010002_

Round 1

Reviewer 1 Report

Work really has a bad title for me.Only physicochemical and qualitative properties are described, but there are no electrochemical tests that impose the title of work. That's why I believe that the mauscript does not prove its purpose at this time. Changing this topic would allow the qualification and description of the properties themselves. That's my biggest complaint

Reviewer 2 Report

In this paper, the authors describe the performances of a new type of separator for batteries made of cellulose nanofibrils prepared by a paper making process. The authors compared the membranes made with 5 different wood in terms of wettability with the electrolyte, thermal stability, morphology and mechanical properties. Compared to Celgard, the membrane tested had a better electrolyte wettability and thermal stability up to 295°C (because Celgard melts partially above 140°C). However, cellulose separator are less interesting in terms of thickness, thermal stability from 295°C to 400°C and mechanical properties (relative to the thickness of the sample). 

It is interesting to look for new separators for batteries, but the tested samples did not surpass Celgards in most important features. Moreover, electrochemical tests are needed to see if the membranes are stable in contact with lithium, that is generally not the case with cellulose at high voltage. It is also important to say that cellulose without purification contains a lot of water residues, that are difficult to evacuate and could not be used in a lithium cell easily. It is thus premature to say that they are "outstanding candidates" as separators.

In particular, I have some remarks about the text : 

-Thermos-dimensional is not a common term, please replace it with thermal. I understand that you want to talk about the shape retention, but it is not clear like this.

-what is the thickness of the membranes that you test? are you sure that it is interesting to have such a thick separator in a lithium cell?

-you compare the tensile stregth of a 20 micron-thick Celgard and a much thicker cellulose separator, wich is not comparable. in general, the mechanical results should be more detailed and explained, with the values reported in a table.

-the morphology section must be put before all other results,and should be used to explain some of the results. figure 8. e1 is missing.

-the results are just described and must contain a more thorough analysis.

-the citations are 100% from asian authors and must be better globally distributed

In general, the english level is poor and the work must be corrected by a native speaker. 

Please correct some errors hand written in the attached file. I did not mark all the mistakes and misformulations in the text, but they should be checked.

Reviewer 3 Report

The authors prepared several cellulose membranes by beating, grinding and homogenizing of natural pulps for using as separators for lithium ion battery. The electrolyte wettability, thermal dimensional stability and tensile strength of these cellulose membranes were characterized. The results are interesting and the manuscript can be published in Materials.

Figure 6 need to be redrawn, and the legend for figure 6 “The (a) DSC and (b) TGA curves of Celard2325 and cellulose membranes before and after the homogenization” should be fixed. For example, as “ The (a) DSC curves of Celard2325 and cellulose membranes without homogenization, and (b) TGA curves of Celard2325 and the homogenized cellulose membranes.

Round 2

Reviewer 1 Report

Can be accepted..

Reviewer 2 Report

The manuscript was extensively revised from version 1 and is now in my opinion ready for publication